# In Vitro/In Vivo Translation of Synergistic Combination of MDM2 and MEK Inhibitors in Melanoma Using PBPK/PD Modelling: Part III

**DOI:** 10.3390/ijms24032239

**Published:** 2023-01-23

**Authors:** Jakub Witkowski, Sebastian Polak, Dariusz Pawelec, Zbigniew Rogulski

**Affiliations:** 1Faculty of Chemistry, University of Warsaw, Pasteura 1, 02-093 Warsaw, Poland; 2Adamed Pharma S.A., Adamkiewicza 6a, 05-152 Czosnów, Poland; 3Faculty of Pharmacy, Jagiellonian University, Medyczna 9, 30-688 Krakow, Poland; 4Simcyp Division, Certara UK Limited, Level 2-Acero, 1 Concourse Way, Sheffield S1 2BJ, UK

**Keywords:** anticancer drugs, virtual clinical trials, pharmacokinetics, pharmacodynamics, drug combination, PBPK/PD modelling, MDM2 inhibitor, MEK inhibitor

## Abstract

The development of in vitro/in vivo translational methods and a clinical trial framework for synergistically acting drug combinations are needed to identify optimal therapeutic conditions with the most effective therapeutic strategies. We performed physiologically based pharmacokinetic–pharmacodynamic (PBPK/PD) modelling and virtual clinical trial simulations for siremadlin, trametinib, and their combination in a virtual representation of melanoma patients. In this study, we built PBPK/PD models based on data from in vitro absorption, distribution, metabolism, and excretion (ADME), and in vivo animals’ pharmacokinetic–pharmacodynamic (PK/PD) and clinical data determined from the literature or estimated by the Simcyp simulator (version V21). The developed PBPK/PD models account for interactions between siremadlin and trametinib at the PK and PD levels. Interaction at the PK level was predicted at the absorption level based on findings from animal studies, whereas PD interaction was based on the in vitro cytotoxicity results. This approach, combined with virtual clinical trials, allowed for the estimation of PK/PD profiles, as well as melanoma patient characteristics in which this therapy may be noninferior to the dabrafenib and trametinib drug combination. PBPK/PD modelling, combined with virtual clinical trial simulation, can be a powerful tool that allows for proper estimation of the clinical effect of the above-mentioned anticancer drug combination based on the results of in vitro studies. This approach based on in vitro/in vivo extrapolation may help in the design of potential clinical trials using siremadlin and trametinib and provide a rationale for their use in patients with melanoma.

## 1. Introduction

Conducting well-designed studies to establish the pharmacokinetic–pharmacodynamic (PK/PD) relationships in animal models in a way that allows us to scale the results to humans is a crucial element of preclinical drug development. Determining and understanding the relationships (or lack of them) between PK and PD significantly improves the interpretation of drug-related data and facilitates successful translation to human conditions [1].

The translational sciences aim to transfer results from basic science to the animal or patient level (bench to bedside). Currently, in vitro/in vivo extrapolation (IVIVE) methods combined with physiologically based pharmacokinetic (PBPK) modelling are often an inherent element of pharmacokinetic (PK) and pharmacodynamic (PD) properties estimation in the drug discovery and development process [2,3,4]. Such an approach can potentially help implement the 3R principles (replacement, reduction, and refinement) for the ethical use of animals by replacing living animals and reduce the number required to obtain meaningful PK/PD data [5].

Despite substantial progress in the development of in silico methods, the challenge of in vitro model verification and in vitro result extrapolation to animals or humans in vivo remains [6]. That is why studies involving animals are still irreplaceable at the preclinical drug development level, especially in the field of oncology [7]. This situation is also caused by the fact that some preclinical results may not directly translate to the clinic because some in vitro systems do not fully mimic the in vivo environment. This is why further development of in vitro/in vivo translational methods is crucial to better characterise a clinically observed drug’s efficacy and safety and to increase the ethical usage of animals and decrease the number of animals needed in preclinical testing programs. One alternative way to avoid the excessive use of animals is the concept of bidirectional translational research. Clinical and in vivo animal studies inform basic science and in vitro research, and vice versa [8]. The translational concept of the bidirectional “learn, confirm and refine” paradigm is adopted in the modelling and simulation (M&S) approach [9]. One of the M&S applications that bridges the gap between in vitro and in vivo research is PBPK/PD modelling [10,11]. The selection of sensitive cancer types usually requires preclinical in vitro and in vivo efficacy data using indication-related cell lines or patient-derived xenografts (PDX). Such a process demands translational studies and data modelling to estimate effective clinical doses and dosing schedules. This is especially important for difficult-to-treat cancer indications requiring drug combinations. The modelling of such data is additionally complicated by the fact that there are a lack of IVIVE solutions for drug combinations. Admittedly, there have been some attempts [12,13]; however, the proposed clinical translational solutions were based only on the in vitro data and neglected the animal in vivo verification context. It is also worth mentioning the concepts of tumour static concentration and tumour static exposure developed by Cardilin et al. [14,15]. This methodology may also aid in translational efforts for drug combinations utilising the Chou–Talalay combination index (CI) theorem [16] and in vivo-derived data. The same group also recently published an interesting scaling technique to predict the clinical efficacy of drug combinations based on their efficacy in mouse xenografts [17].

Metastatic melanoma is a cancer condition that is life-threatening and difficult to treat due to its ability to spread early and aggressively. The treatment of aggressive and fast-spreading cancer usually requires a combination of various therapeutic options to stop the cancer from developing further. One of the newly proposed therapeutic options is the drug combination of siremadlin MDM2 (mouse double minute 2) inhibitor and trametinib MEK (mitogen-activated protein kinase kinase) inhibitor. Drug combinations utilising this class of inhibitor are currently the subject of many studies in clinical trials (clinicaltrials.gov (accessed on 14 December 2022) identifiers: NCT02110355, NCT03714958, NCT02016729, NCT01985191 and NCT03566485). Preclinical evidence suggests that siremadlin (previously known as HDM201) and trametinib act synergistically in melanoma [18,19,20,21,22]. The mechanisms of action of both drugs (including the mechanisms influencing melanin production [23,24], which may prevent melanoma metastasis [25,26]) are depicted in Appendix A.

Moreover, previous in vitro and in vivo data suggest that these two compounds might synergistically interact with each other at the PK and PD levels; thus, co-administration may result in synergistic drug–drug interaction (DDI) at the PK and PD levels, which provides the basics for further consideration as a therapeutic option.

Recent studies showed that a tumour size change of at least 10% is a prognostic factor and is highly correlated with the improvement of overall survival in melanoma [27,28]. Thus, it is believed that the increased antitumour activity of the siremadlin and trametinib drug combination may contribute to increased overall survival of melanoma patients.

The main aim of this study was to establish an accurate simulation methodology for drug combination using available in vitro/in vivo absorption, distribution, metabolism, and excretion (ADME), and PK and PD data, to develop and optimise a PBPK/PD model. Such a model could potentially allow for the translation of previously presented in vitro/in vivo findings into an in vivo human situation. This model was used to estimate the clinical effect (tumour size reduction) of an MDM2 and MEK inhibitor combination.

The performance of virtual clinical trials (VCTs), which might be carried out on a virtual representation of cancer patients, may lead to accurate estimations of the drug combination’s efficacy. This can ultimately provide the rationale for using siremadlin and trametinib in combination in clinical trials with melanoma cancer patients.

## 2. Results

### 2.1. PBPK Models (with and without PK Interaction)

The two PBPK models were developed for siremadlin. The first model with a mixed zero- and first-order absorption mechanism (hereafter referred to as the MO model) was originally proposed in the literature [29], and the second one was simplified to a first-order absorption mechanism only (referred to as the FO model). Both models reasonably well described the plasma concentration–time data for siremadlin administered daily and in the intermittent regimens in cancer patients’ representatives, as shown in the example in Figure 1 and Appendix A. PBPK simulations for siremadlin for all the study participants are depicted in Appendix A.

The PBPK models developed for siremadlin indicated that the estimated area under the curve (AUC_0-inf_, hereafter also referred to as exposure) and maximal concentration (C_max_) values were accurately predicted only with the FO model. The numerical analysis of errors (fold difference between the predicted and clinically observed values) for the most important pharmacokinetic parameters, AUC_0-inf_ and C_max_*,* showed that values predicted by the FO model were within the 2-fold error range (0.5–2.0) of the observed values, except the exposure predicted for a 4 mg dose (Table 1 and Table 2). The exception for a 4 mg dose was most probably caused by the fact that the observed exposure for a 4 mg dose was not dose-proportional. The observed exposure ratio at 4 mg and 2 mg doses (AUC 4 mg dose/AUC 2 mg dose) was 1.27, whereas the predicted exposure ratio at those doses was dose-proportional (the AUC ratio equalled 1.96). Further analyses also showed that the fold errors of AUC_0-inf_ and C_max_ predicted by the FO model were closer to unity compared to those predicted by the MO model (Appendix A). Moreover, the introduction of the FO model decreased over a 50× computation time and allowed for model simplification with increased precision. This is why the first-order model was selected as the absorption mechanism in the final PBPK model and was further used in the estimation of PK interaction and in drug combination.

The PBPK model developed for trametinib properly described the plasma concentration–time profiles in cancer patients after a single 2 mg dose and multiple (15) doses, as shown in Figure 2 and Figure 3. The PBPK simulations for trametinib for all study participants are depicted in Appendix A.

The numerical analysis of errors (fold difference between the predicted and observed values) for the most important pharmacokinetic parameters, AUC_0–24 h_ and C_max_, showed that the predicted data met the 2-fold error (range: 0.5–2.0) acceptance criteria (Table 3).

The last step of PBPK model development for siremadlin and trametinib was the estimation of possible PK interaction at the absorption level using the AUC ratio relationships established previously in animals [19]. Plasma concentration–time profiles for siremadlin and trametinib with and without estimated PK interaction are shown in Figure 4 and Figure 5.

The predicted changes in AUC_0–24 h_ and C_max_ for siremadlin and trametinib with and without PK interaction for population representative are shown in Table 4. The final siremadlin and trametinib PBPK models’ input parameters with and without PK interactions are summarised in Appendix A.

### 2.2. PD (TGI) Models

The final perturbed tumour growth inhibition (TGI) model for siremadlin and trametinib assumed exponential tumour growth, the Skipper–Schabel–Wilcox (log-kill) tumour-cell-killing hypothesis, the drug effect described by the linear drug-killing model, primary resistance to the therapy, and treatment effect delay described by the signal distribution model with four transit compartments. Changes in the sum of the longest diameters (SLD, hereafter also referred to as tumour size) over time in cancer population representatives after treatment with siremadlin (various doses and dosing regimens) and trametinib (2 mg dose administered daily) are presented as examples in Figure 6, Figure 7, Appendix A. TGI models for siremadlin and trametinib administered in monotherapy in the cancer patient population are depicted in Appendix A. Since there was no available information about tumour growth in the trametinib study, two TGI model variants were developed. The first one had high tumour growth of kgh = 0.00028 1/h (Figure 8) and low tumour growth of kgh = 0.0000261 1/h (Figure 9). For drugs administered in monotherapy, this model was able to accurately capture the changes in tumour size over time, with a mean relative error (RE) < 20% (except for siremadlin administered in regimen 1A with a 25 mg dose, which was slightly over the threshold—20.06%), as shown in Appendix A. The key input parameters (with corresponding %CV calculated based on equations from [33]) for the final TGI models for siremadlin and trametinib monotherapy are shown in Appendix A.

### 2.3. PBPK/PD Simulations for Drug Combination

The drug combination model was built based on universal TGI model principles previously established in animals [19] and a developed TGI model for siremadlin and trametinib. The relationships between TGI parameters for the drug combination are shown in Appendix A. For the drug combination simulations, siremadlin’s recommended dose for expansion of 120 mg (in regimen 1B) [36] and an approved dose of 2 mg (daily dosing) [37] of trametinib were selected. Several scenarios were simulated for the combination of siremadlin and trametinib, including the assumption that the drugs will interact with each other at the PK or PD level. Additionally, several parameters were tested to examine how they influence the combination treatment outcome:Case A assumed high tumour growth (kgh), high initial tumour size (SLD0), and a high fraction of sensitive cells.Case B assumed a similar approach to that described above, but assumed low tumour growth (kgh).Case C assumed high tumour growth (kgh), high initial tumour size (SLD0), and a low fraction of sensitive cells.Case D assumed high tumour growth (kgh), low initial tumour size (SLD0), and a high fraction of sensitive cells.

Several sub-scenarios were also included in which the potential influence of interactions at the PK and PD levels between siremadlin and trametinib were studied:Scenario 1—without PK and PD drug interactionsScenario 2—without PK but with PD drug interactionsScenario 3—with PK and without PD drug interactionsScenario 4—with PK and PD drug interactions

The previously proposed universal TGI model for the siremadlin and trametinib drug combination allowed us to extrapolate the values of the clinical model parameters in a melanoma cancer population. The key input parameters, relationships between input parameters, and simulation results for these scenarios are shown in Figure 10, Figure 11, Figure 12 and Figure 13 and Appendix A.

The simulations of tumour growth patterns after administration of the siremadlin and trametinib combination indicated that there was a substantial tumour size reduction, regardless of the occurrence of PK or PD interaction. The simulations of tumour growth pointed out a biphasic response to the drug combination with initial and subsequent delayed tumour size reduction (cases A and D), while a monophasic response was observed with low tumour growth and a low fraction of sensitive cells (cases B and C). The simultaneous introduction of PK and PD interactions had the greatest impact on the tumour growth reduction (scenario 4) while a lack of such interactions resulted in only a limited increase in tumour growth reduction over trametinib monotherapy (scenario 1). The introduction of interaction only at the PD level had a lower effect than the introduction of interaction at the PK level in terms of tumour size reduction (scenarios 2 and 3, respectively). The simulated tumour growth dynamics affected the final tumour size (size at the end of the 30-week simulation: 5040 h) for the most efficacious scenario in which PK and PD interactions were employed. For highly proliferating tumours, the simulated neoplasm size was reduced to around 1 cm, while for low tumour growth, it was approximately four-times higher (4 cm).

The increased combination efficacy as a mean objective response rate (ORR—percentage of patients with a complete or partial response at any time during the study) was statistically significant in each drug combination scenario compared to the most efficacious drug—trametinib (Table 5).

## 3. Discussion

The developed PBPK/PD models for the MDM2 inhibitor siremadlin and the MEK inhibitor trametinib were able to describe both the pharmacokinetic and pharmacodynamic profiles of these drugs. The models consider the oral (p.o.) administration of siremadlin and trametinib, the first-order absorption mechanism, a full-body distribution model, hepatic metabolism and renal excretion for siremadlin (renal excretion calculated using the FCIM method [38]), intravenous clearance for trametinib, and a permeability-limited tumour distribution model.

The first developed PBPK model for siremadlin assumes a mixed zero- and first-order absorption mechanism, as originally suggested in the literature [29], and the second model includes a first-order absorption mechanism only. The analysis of the fold difference between the predicted and clinically observed values for key PK parameters (AUC_0-inf_ and C_max_) indicated that those values were accurately predicted only with the model assuming a first-order absorption mechanism. In our opinion, the model with mixed zero- and first-order absorption is more sensitive to the initial tumour size (SLD0) than the model with first-order absorption. Therefore, the predicted values may be more likely to deviate from those observed in the clinical trial. The exposure and maximal concentration values predicted by the model assuming first-order absorption were within the 2-fold error range (0.5–2.0) of the observed values, except the AUC_0-inf_ predicted for a 4 mg dose. The exception for the 4 mg dose was most probably caused by the fact that the observed exposure for a 4 mg dose was not dose-proportional compared, for example, with exposure for a 2 mg dose (AUC 4 mg dose/AUC 2 mg dose: 1.27), whereas the predicted exposure was nearly dose-proportional (AUC 4 mg dose/AUC 2 mg dose: 1.96). The model tended to slightly overpredict exposure in the 2–25 mg dose range, while for higher doses, the fold error was close to unity. This may have been caused by several factors: different sampling of data in the simulation and in the clinical protocol, or an unexplained variability of nearly 17%, which was assumed during the generation of resimulated PK data [29], while in the PBPK model, no unexplained PK variability was assumed. Since the PBPK models consider various compartments, rapid changes in tumour size may affect the volume of distribution, which could also explain the variability in the estimated PK parameters. The final PBPK model was successfully verified with resimulated data from the PK model published by Guerreiro et al. [29] and external PK data digitised from the literature [39].

The PBPK model developed for trametinib properly described the plasma concentration–time profiles in cancer patients after a 2 mg dose administered once and after reaching a steady state (after approximately 15 doses [40]). The predicted data met the 2-fold error range (0.5–2.0) acceptance criteria for AUC_0–24 h_ and C_max_, as well. The developed PBPK model was successfully verified with external PK data extracted from the literature [30,31,32].

The above-mentioned models were further extended with the introduction of PK interaction at the absorption level. Estimations of the exposure changes for both drugs were made based on relationships between the AUC ratios and drug doses previously established in animals [19]. This interaction included changes in the absorption rate constant (ka) and fraction absorbed (fa) for siremadlin and alterations in the absorption rate constant (ka), fraction absorbed (fa), and lag time (tlag) in the case of trametinib.

The translation of PK drug interaction from animals to patients is very challenging. This is due to differences in digestive tract anatomy and physiology, transporter abundance, intestinal pH, internal organ blood flow, metabolising enzymes, and potential drug absorption mechanisms [41,42]. It appears that ex vivo or cadaver studies in an Ussing chamber might be one of the methods that allows for precise estimation of the fraction absorbed for single drugs and drug combinations. However, further studies are needed to prove its utility [43,44,45]. Therefore, the occurrence of PK interaction between siremadlin and trametinib estimated in this work should be taken with caution. Only results from the currently ongoing siremadlin and trametinib clinical trial (NCT03714958) or following trials may confirm or exclude the occurrence of such PK interaction.

Previously performed in vitro and in vivo studies have shown that the killing effect of siremadlin and trametinib compounds is concentration- and time-dependent, and an initial delay in the response and resistance to the treatment might arise [18,19]. These findings allowed us to develop a perturbed TGI model for siremadlin and trametinib. The developed model assumed exponential tumour growth, the Skipper–Schabel–Wilcox (log-kill) tumour-cell-killing hypothesis, the drug effect described by the linear drug-killing model, and acquired resistance to the therapy and treatment effect delay described by the signal distribution model with four transit compartments. The delay in the effect of these drugs is most likely related to the duration of the signal transduction associated with the activation of the p53–MDM2 and MAPK pathways, resulting in cell death. Resistance is an inherent part of anticancer treatment; therefore, its description may play a critical role in predicting and optimising the treatment response and may improve therapy scheduling [46,47]. Clinically predicted resistance in siremadlin and trametinib models was much higher (13–26× higher) than estimated in animal models, which further warrants the need for drug combination usage to overcome it. The applied model assumed that two distinct tumour growth patterns might occur as originally proposed for solid tumours [29], the first with high tumour growth 0.00028 cm/h and the second with low tumour growth 0.0000261 cm/h. These assumptions were in line with the reported tumour growth range observed in melanoma cancer (0.00030 cm/h and 0.000015 cm/h for high and low melanoma growth, respectively [48]) and a previously performed study with melanoma-derived A375 xenografts in mice (0.00024–0.00031 1/h range [19]). Therefore, 0.00028 cm/h and 0.0000261 cm/h were used as estimates for high and low tumour growth in the melanoma population, as well.

The final TGI models for the perturbed groups properly predicted the tumour size within 20% of the mean relative error (%RE) between the predicted and resimulated data in the case of siremadlin (except for a 25 mg dose of siremadlin administered in regimen 1A), and the predicted versus the literature data for trametinib. Most of the observed differences were related to the TGI model developed for siremadlin. This may have been due to the fact that when generating the resimulated PD data, nearly 4% of unexplained variability [29] was assumed in contrast to the PBPK/PD model, where no unexplained PD variability was assumed. Moreover, differences in the predicted PK between the PK model in Simulx and the PBPK model developed in Simcyp may additionally deepen these differences. We believe that further modelling work is warranted to optimise this TGI model to ensure that all doses and dosing regimens meet the 20% acceptance criteria.

No clinical data from the siremadlin and trametinib combination are available yet. Simulation of this combination required us to test virtual “what if” scenarios related to the influence of various model parameters and the introduction of potential interactions at the PK and PD levels. The results of an in-depth analysis of the TGI model parameter dependencies and experiences from mouse species allowed us to extrapolate PD model predictions for this combination into a clinical context. The developed PK interaction model for siremadlin and trametinib was used in simulations, assuming PK interaction during co-administration and that the previously selected β parameter from the MuSyC drug interaction model was used as the PD interaction parameter.

The simulations covered the influence of the tumour growth pattern (kgh parameter), fraction of sensitive cells (fs parameter) and initial tumour size (SLD0 parameter) on the siremadlin recommended dose for expansion of 120 mg dose (in regimen 1B) [36] and the approved 2 mg dose (daily dosing) [37] of trametinib. As mentioned earlier, two distinct tumour growth estimates (kgh high and kgh low), which correspond to the values observed in human melanoma [48], were tested in simulations of efficacy in patients with melanoma. The fraction of sensitive cells for the drug combination was unknown; thus, two assumptions were made: first, that the estimated sensitive fractions will be summed up for drug combination (fs high), and second, that there will only be some parts of the cells that respond (fs low). The assumed low fraction of sensitive cells was equal to the inter-individual variability of the fs parameter for siremadlin (0.0321 × 1.93 = 0.0620), which was similar to the reported values for pembrolizumab in melanoma (0.0628) [49]. Since no clinical data are available, these assumptions should also be taken with caution. Moreover, this model did not assume conversion from sensitive to resistant cells, which might occur in the clinical trials [50,51,52,53]. However, this issue is out of the scope of this work because access to raw clinical data would be needed to verify such a modelling hypothesis. According to the pooled analysis from four randomised clinical trials [54], the baseline SLD value was a prognostic factor of the overall survival of patients treated with MEK and BRAF inhibitors if the initial SLD (SLD0) was lower than or equal to 4.4 cm; therefore, the influence of this parameter on the combination’s efficacy was also tested (Case D).

The simulations for the drug combination showed that high tumour growth (khg high) results in a biphasic response to the drug combination, with an initial and subsequent delayed tumour size reduction, regardless of the initial tumour size (SLD0), as shown in cases A and D. In turn, a monophasic response was observed in simulations with low tumour growth (kgh low) and a low fraction of sensitive cells (fs low), as may be observed in cases B and C. These simulation results suggest that the tumour characteristics (such as tumour growth or sensitivity to combined drugs) may have a considerable impact on the onset of the observable therapeutic effect. The simultaneous introduction of PK and PD interactions had the greatest impact on tumour growth reduction (scenario 4), while a lack of such interactions resulted in only a limited increase in tumour growth reduction over trametinib monotherapy (scenario 1). This may be simply explained by the fact that all predicted PK and PD interactions were acting synergistically, amplifying the predicted response. The introduction of the interaction only at the PD level had a lower effect than the introduction of the interaction at the PK level in terms of tumour size reduction (scenarios 2 and 3, respectively). The simulated tumour growth dynamics affected the final tumour size (size at the end of the 30-week (5040 h) simulation) for the most efficacious scenario in which PK and PD interactions were employed. For highly proliferating tumours (kgh high), the simulated neoplasm size was reduced to ~1 cm, while for low tumour growth (kgh low), it was four-times higher (~4 cm). Interestingly, alteration of the initial tumour size (Case D) did not increase the calculated ORR; however, further modelling with different doses and dosing regimens is needed to confirm such an observation.

The simulation results indicated that this drug combination significantly increases the probability of achieving at least a partial response in melanoma patients regardless of the occurrence of PK and PD interaction. For Cases A, C and D, the calculated ORR was in the 61–98% range, which seems to be comparable to the reported ORR for the already approved dabrafenib (BRAF inhibitor) and trametinib combination (ORR range: 50–76% [55,56,57,58,59]). Nevertheless, the estimation of the superiority or noninferiority of the siremadlin and trametinib combination over the already approved dabrafenib and trametinib combination was not possible with such limited availability of data. Hence, without access to raw PK/PD clinical data coupled with the development of population PBPK models for those molecules, such conclusions cannot be made.

The calculated ORR for trametinib monotherapy in Case B was twice as low as in other scenarios, and was the most comparable with the observed ORR in the clinic (26 vs. 22%) [35]. Likewise, in Case B for drug combination scenarios, a lower ORR was calculated (ORR range: 48–81%), which also seems to be comparable with the data reported for the approved dabrafenib and trametinib combination. This finding supports the hypothesis that all patients suffering from metastatic melanoma, regardless of baseline tumour size, tumour growth dynamics, and the fraction of sensitive cells, may potentially benefit from this drug combination therapy. It should also be noted that other clinical factors can affect the clinical response assessment, such as shrinkage in nontarget tumours such as pathologic lymph nodes or the appearance of malignant lesions indicating cancer progression. Due to the fact that the developed models accounted only for the tumour size of the target lesion without a distinction between lymph nodes or tumour metastases, such issues described in the Response Evaluation Criteria in Solid Tumors (RECIST) guidelines [60] cannot be fully accounted for in such simulations. Therefore, the interpretation of results and drawing of direct comparisons with the actual clinical response should always be approached with caution.

The developed PBPK/PD models also have other associated limitations which should be discussed. Synergistic pharmacodynamic interaction in the current TGI model was based on results from in vitro studies that were further validated on mouse xenografts from the same A375 melanoma cells. One of the limitations of such an approach is that cell-line-derived xenograft models cannot fully simulate the microenvironment of tumours in humans, such as the vascular, lymphatic, and immune environments; moreover, such models often lose genetic heterogeneity compared to primary tumours [61]. In recent years, the patient-derived xenograft model (PDX model) has emerged as a promising tool that provides translational value with better mimicking of the tumour microenvironment; however, one of the disadvantages of this model is the loss of the human tumour stroma, which is entirely replaced by the murine stroma [62,63]. There is evidence that both the tumour stroma and the corresponding microenvironment may affect the drug response [64,65,66]. This may be important based on the reports that MDM2 inhibition may affect stromal or immune microenvironments, which was not considered in this work [67,68,69]. The results from ongoing clinical trials (NCT03611868, NCT03787602) may shed light on the utility of MDM2 inhibitors in skin cancer.

Another troubling problem regarding the developed PBPK/PD model is the repetitiveness of the pharmacodynamic interaction parameter (β) in the heterogeneous melanoma patient population. Additional in vitro studies, followed by in vivo verification with melanoma-derived xenografts or patient-derived xenografts (PDX), are needed to validate the magnitude of the estimated pharmacodynamic synergy between siremadlin and trametinib and to improve the translational value of the present study.

The current TGI model did not account for the cell transition from a sensitive to a resistant state (defined in the literature as the ksr parameter [29,51]), which might be useful for the estimation of long-term therapies such as cancer treatment. Further model development with raw clinical study data would be needed to assess whether the introduction of such a feature would improve the fit to clinically observed data for siremadlin and trametinib.

Due to the limited availability of response data for trametinib in the literature, the developed model may be not optimal and does not cover delayed resistance, resulting in tumour recurrence in patients with partials and complete responses. Thereby, the developed TGI model for trametinib predicts a higher percentage of responders than clinically observed (especially in the scenario in which high tumour growth is assumed) [35,70,71].

Nonetheless, despite these many limitations, the developed PBPK/PD models reasonably accurately described the PK and time course of tumour growth across all doses and dosing schedules. Further analyses and modelling work with different doses and dose regimens are encouraged to externally validate the developed PBPK/PD models for siremadlin, trametinib, and their combination toward predicting tumour size reduction. Although modelling work with different doses and dosing schedules is still ongoing, the results from the in vitro/in vivo translational approach presented in this study are promising. This study shows that the currently examined 120 mg dose of siremadlin in regimen 1B with daily dosed trametinib may act with sufficient synergy to elicit a significant clinical response.

Further performance of virtual clinical trial simulations with different doses and dosing regimens of the siremadlin and trametinib combination are ongoing to help select the most synergistic, efficacious, and safe dose levels and dosing regimens for melanoma-bearing patients. Further development of the presented clinical PBPK/PD models for siremadlin and trametinib is needed to develop a proper drug combination model in clinical settings and aid in the design of potential clinical trials using siremadlin and trametinib in patients with melanoma.

## 4. Materials and Methods

### 4.1. Clinical Studies Used

Data modelling usually requires two types of dataset: a training dataset, used in the development of the model, and a verification dataset, used for independent verification of the developed model. When early clinical data are available (e.g., from phase I clinical studies) a PK model is developed using only data from such studies and validated using data from later phases of clinical studies (phases 2 and 3). For siremadlin, resimulated data from Guerreiro et al. [29] were used as a training dataset and data from Stein et al. [29] and Jeay et al. [39] as verification datasets.

The details of data from clinical studies used for PBPK and PD (TGI) model development and verification are summarised in Table 6.

### 4.2. Software

The PK parameters were estimated using Microsoft Excel (Excel version 2016, Microsoft Corporation, Redmond, WA, USA, 2016, https://www.office.com (accessed on 14 December 2022)). Digitisation of the literature-derived data was performed with the use of WebPlotDigitizer software (version 4.4, Ankit Rohatgi, Pacifica, CA, USA, 2021, https://automeris.io/WebPlotDigitizer (accessed on 14 December 2022)). TGI modelling for trametinib was performed using Monolix software (Monolix version 2021R1, Lixoft SAS, a Simulations Plus company, Antony, France, 2022, http://lixoft.com/products/monolix/ (accessed on 14 December 2022)). The simulation of clinical PK/PD data for siremadlin was performed using Simulx software (Simulx version 2021R1, Lixoft SAS, a Simulations Plus company, Antony, France, 2022, https://lixoft.com/products/simulx/ (accessed on 14 December 2022)).

The Mlxtran model used for PK/PD resimulation in Simulx can be found in Appendix A. PBPK/PD modelling was performed using Simcyp simulator software (Simcyp simulator V21, Certara UK Limited, Sheffield, UK, 2022, https://www.certara.com/software/simcyp-pbpk (accessed on 14 December 2022)). The custom mixed zero- and first-order absorption PK for siremadlin, PK drug interaction for trametinib, and PD (TGI) models in Lua can be found in Appendix A. Fisher’s test was performed using GraphPad Prism version 9.4.1 for Windows, GraphPad Software, San Diego, CA, USA, 2022, www.graphpad.com (accessed on 14 December 2022).

### 4.3. Statistical Methods

According to RECIST v1.1 metric [60], patients whose calculated maximal % reduction in change from baseline tumour size was ≥30% at any time during study were scored as responders (PR—partial response and CR—complete response), and the others as non-responders (SD—stable disease and PD—progressive disease). The percentage of responders represents the overall response rate (ORR). The ORRs were tabulated based on the number and percentage of subjects attaining an overall best response of complete or partial response in the melanoma patient population. Fisher’s exact test was used to evaluate the ORR statistically between trametinib and the estimated trametinib plus siremadlin drug combination.

From the available sample size—*n* = 457: 214 patients from the trametinib study and 243 patients (214 patients from the trametinib study plus 29 patients from the siremadlin study) from the trametinib/siremadlin drug combination study—we calculated the precision of the effect (%ORR) estimate between trametinib and trametinib plus siremadlin to be 12.7% at a statistical power corresponding to 80% using a two-sided test with a significance level of α = 0.05. The precision of the effect was calculated using a sample size calculator [72]. Microsoft Excel was used to calculate the %ORR and GraphPad Prism was used to perform Fisher’s test.

### 4.4. Resimulation of Clinical PK and PD Data for Siremadlin

In the literature, PK profiles for siremadlin were available only for 46 individuals [39] instead of the whole cohort involved in clinical trial (*n* = 115). Additionally for already published PD data [29], the assignment of particular dosing levels and dosing schedules was not possible. For these reasons, patients’ siremadlin PK and PD profiles were resimulated in Simulx using the PK/PD model from [29].

The PK and PD model input parameters used for resimulation in Simulx are shown in Appendix A. Descriptions of the dose levels, dosing schedules, and the number of patients in each resimulated group are shown in Table 7.

**Table 7 ijms-24-02239-t007:** Descriptions of resimulated groups treated with siremadlin.

Dose (mg)	Regimen	Dosing Schedule	No. of Patients *	No. of Trials	Notes
1	2A	qdx14 in 28-day cycle	1	10	
2	2A	qdx14 in 28-day cycle	2	10	
4	2A	qdx14 in 28-day cycle	4	10	
7.5	2A	qdx14 in 28-day cycle	4	10	
15	2A	qdx14 in 28-day cycle	4	10	
20	2A	qdx14 in 28-day cycle	5	10	
15	2C	qdx7 in 28-day cycle	8	10	
20	2C	qdx7 in 28-day cycle	6	10	
25	2C	qdx7 in 28-day cycle	5	10	
12.5	1A	qdx1 in 21-day cycle	1	10	
25	1A	qdx1 in 21-day cycle	1	10	
50	1A	qdx1 in 21-day cycle	4	10	
100	1A	qdx1 in 21-day cycle	4	10	
200	1A	qdx1 in 21-day cycle	5	10	
250	1A	qdx1 in 21-day cycle	9	10	Including patients from eltrombopag group (*n* = 3)
350	1A	qdx1 in 21-day cycle	5	10	
120	1B	qwx2 (day 1/8) in 28-day cycle	29	10	
150	1B	qwx2 (day 1/8) in 28-day cycle	15	10	Including patients from eltrombopag group (*n* = 7)
200	1B	qwx2 (day 1/8) in 28-day cycle	3	10	

* Numbers of patients were obtained from siremadlin clinical trial protocol [73].

### 4.5. Physiologically Based Pharmacokinetic Models

#### 4.5.1. General PBPK Modelling Strategy

The modelling strategy was based on the “middle-out” approach combining the advantages of the “bottom-up” and “top-down” approaches [74]. In our case, some parameters were fixed (such as in vitro-determined or literature-derived data for siremadlin and trametinib [18,19,29,37,39,40,75,76,77,78,79,80]) and others were estimated. Parameter estimation was performed using the PE Module of the Simcyp Simulator V21 using the Nelder–Mead method, the weighted least squares by the reciprocal of the square of the maximum observed value as the objective function, and a termination criterion defined as improvement of less than 1% of the objective function value. Optimisation was performed manually to fit the observed or resimulated data in the case of siremadlin. PBPK model performance was evaluated based on the “2-fold” criterion for maximum concentration (C_max_) and area under the concentration–time curve (AUC) [81,82].

#### 4.5.2. Virtual Population Characteristics (System Data)/Patient Population

As both studied drugs are intended for use in patients with cancer, the standard cancer population (Sim-Cancer) which is included within the Simcyp simulator was used. This software allows us to perform simulations on a typical cancer population patient (population representative) or on the whole cancer population. This special population has many adjustments (age distribution, height–age–weight relationships, prediction of glomerular filtration rate, and changes in plasma protein concentrations, e.g., alpha-1-acid glycoprotein, human serum albumin, and serum creatinine), which are made to better account for the specific changes that are expected to be found in the physiological parameters of such a population [83,84]. Some of the physiological parameters of the Simcyp cancer population were modified to specifically mimic the melanoma cancer population. Concerning this, tumour properties including tumour tissue size, blood flow (blood flow was set as in human melanoma xenografts [75,79,80]), composition (skin-derived), and pH were adapted to the studied population.

#### 4.5.3. PBPK Model Verification

The final siremadlin and trametinib PBPK models were compared with external observed or resimulated PK data [30,31,32,36,39] through both visual checks and numerical analysis. Predicted longitudinal plasma concentration profiles were generated, including the geometric mean predicted concentrations. Local sensitivity analysis (parameter scanning) was performed to evaluate the relative impacts of fa, ka, tlag, Hep intrinsic CL, fu_inc, and additional renal CL on the plasma PK parameters (AUC_0–24 h_/AUC_0-inf_ and C_max_) for siremadlin, and fa, ka, tlag, and CL_iv_ for trametinib. The performance of the siremadlin and trametinib PBPK models was assessed using the fold error in plasma, which referred to the ratio of the predicted AUC_0–24 h_/AUC_0-inf_ and C_max_ to the observed values (Equation (1)). AUC_0–24 h_/AUC_0-inf_ was calculated using the linear trapezoidal rule. Both visual checks and numerical analyses were performed in Microsoft Excel 2016. Changes in PBPK parameters related to PK DDI at the absorption level are shown in Appendix A.
Fold Error PK parameter = Predicted PK parameter/Observed PK parameter,(1)

### 4.6. Pharmacodynamic Modelling

#### 4.6.1. General PD (TGI) Modelling Strategy

The initial PD model used further in PBPK/PD modelling for siremadlin and trametinib was established using Monolix software. Model selection was based on visual inspection of the individual observed vs. predicted data and a comparison of the resulting values of the model score (Equation (2)).
Model score = −2 × log-likelihood (−2LL, called also objective function value—OFV) + corrected Bayesian Information Criteria (BICc),(2)

−2LL and BICc were estimated using the linearisation method to accelerate the calculations. For the TGI model further developed in the Simcyp simulator, the goodness of TGI model fit was evaluated based on the mean relative error (RE) value (Equation (3)) being < 20%, as proposed in [85]:RE (%) = 100 × (Predicted Tumour Volume − Observed Tumour Volume)/Observed Tumour Volume,(3)

#### 4.6.2. PD (TGI) Model Development and Verification

The TGI model for siremadlin and trametinib was developed based on a model previously proposed by Guerreiro et al. [29]. In this model, unperturbed tumour growth is assumed to be exponential, as proposed for melanoma [86] (Equation (4)):dTS/dt = kgh × SLD0,(4)
where kgh is tumour growth (1/h) and SLD0 is the initial sum of the longest diameters (cm).

After the characterisation of tumour growth, the tumour growth inhibition models (perturbed models) were developed for siremadlin, trametinib, and their combination based on the observed tumour size for trametinib and the resimulated tumour size data for siremadlin.

The final selected perturbed TGI model characterising single siremadlin-, single trametinib-, and combination-treated groups is written in the following equations and under the initial conditions:TotalSLD(t) = Ts(t) + Tsr(t),(5)
Ts_0 = SLD0 × fs,(6)
Tsr_0 = SLD0 × (1 − fs),(7)
TSCs_combination = (TSCs_siremadlin/AUC_ratio_siremadlin—TSCs_trametinib/AUC_ratio_trametinib)/gamma,(8)
TSCs = TSCs_siremadlin|TSCs_trametinib|TSCs_combination,(9)
C(t) = C_siremadlin|C_trametinib|(C_siremadlin + C_trametinib),(10)
TK(t) = kgh/TSCs × C(t),(11)
K1_0 = 0,(12)
K2_0 = 0,(13)
K3_0 = 0,(14)
K4_0 = 0,(15)
dK1/dt = (dTK − K1)/tau,(16)
dK2/dt = (K1 − K2)/tau,(17)
dK3/dt = (K2 − K3)/tau,(18)
dK4/dt = (K3 − K4)/tau,(19)
dTs/dt = 0,(20)
dTsr/dt = 0,(21)
dTs/dt = kgh × Ts − K4 × Ts,(22)
dTsr/dt = kgh × Tsr − K4/(1 + lambda) × Tsr,(23)
where pre-existing resistance is defined by the introduction of the sensitive (Ts—tumour size) and resistant population (Tsr—resistant tumour size) of cancer cells. It was assumed that at time 0, the total sum of the longest diameters of the tumour (TotalSLD) is represented by the sum of Ts and Tsr (Equation (5)). The initial Ts and Tsr size at time = 0 are calculated with the fraction of sensitive cells (fs) and the initial sum of the longest diameters (SLD0), as in Equations (6) and (7). The tumour static concentration for the sensitive fraction (TSCs) for the combination (TSCs_combination) was characterised as the subtraction of the siremadlin and trametinib TSCs adjusted with the PK interaction parameters (the AUC ratio parameters which were calculated for siremadlin and trametinib based on findings from the preclinical model (see Appendix A)) as well as gamma (the PD interaction parameter) and β parameter determined from the analysis of in vitro data [18] (Equation (8)). Depending on the treated group, the TSCs constant could be assigned to the TSCs constants of siremadlin, trametinib, or their combination (Equation (9)). The total plasma concentration of siremadlin, trametinib, or their combination was used as the input for the drug effect (Equation (10)). The tumour-killing effect (TK) assumes linear dynamics of the treatment effect, which is defined by tumour growth (kgh), tumour static concentration for sensitive cells (TSCs), and total plasma concentration (C) (Equation (11)). A delay of the killing effect (TK) was implemented through the introduction of 4 signal-transit compartments (K1, K2, K3, and K4), as suggested by [87]. The duration of this delay is determined by the parameter tau (Equations (16)–(19)). It was assumed that transit compartments equalled 0 at time = 0 (Equations (12)–(15)). It was also assumed that the initial change in tumour size for sensitive and resistant cell populations was equal to 0 (Equations (20) and (21)). The tumour growth rate (kgh) was assumed to be the same for the sensitive and treatment-resistant cell populations. The Skipper–Schabel–Wilcox log-kill hypothesis was applied to describe tumour size changes over time for sensitive and resistant cell populations (Equations (22) and (23)). Additionally, it was assumed that the studied drugs also induce a killing effect on the resistant cell population but with reduced potency (Equation (23)). The parameter lambda denotes the fold-change loss in drug potency in resistant cells relative to sensitive cells. The units for particular parameters are summarised in Appendix A.

In the last step, unperturbed and perturbed tumour growth inhibition models for siremadlin, trametinib, and their combination, previously developed in Monolix, were translated into Lua programming language and applied within the Simcyp simulator V21 for further development to achieve a mean relative error (RE) value of <20% (Equation (3)).

#### 4.6.3. Tumour Size Simulation for the Drug Combination

Tumour size for the studied drugs and their combination was estimated using the PBPK/PD model within a 0–5040 h (210 days) simulation timeframe at clinically relevant doses (120 mg for siremadlin and 2 mg for trametinib) and dosing regimens for each drug (qwx2 for siremadlin and qd for trametinib). The trial design assumed a cancer population in a 20–80-year age range, 50% of patients of either sex, and tumour blood flow typical for melanoma. For simulation purposes, the influence of the initial (baseline) sum of the longest diameters (SLD0) was tested. Each simulated patient response was classified according to RECIST criteria [60]; therefore, all patients whose tumour size decreased by at least 30% were classified as responders. The overall response rate (defined as the percentage of subjects who achieved CR or PR at any time during the study) was calculated for each simulated scenario: drugs administered in monotherapy and drug combinations.

Additionally, according to the RECIST guidelines, the minimum measurable tumour size is 10 mm; therefore, if a patient’s tumour size was lower than 1 cm during the simulation timeframe (5040 h), that patient was also counted as a responding patient.

## Figures and Tables

**Figure 1 ijms-24-02239-f001:**
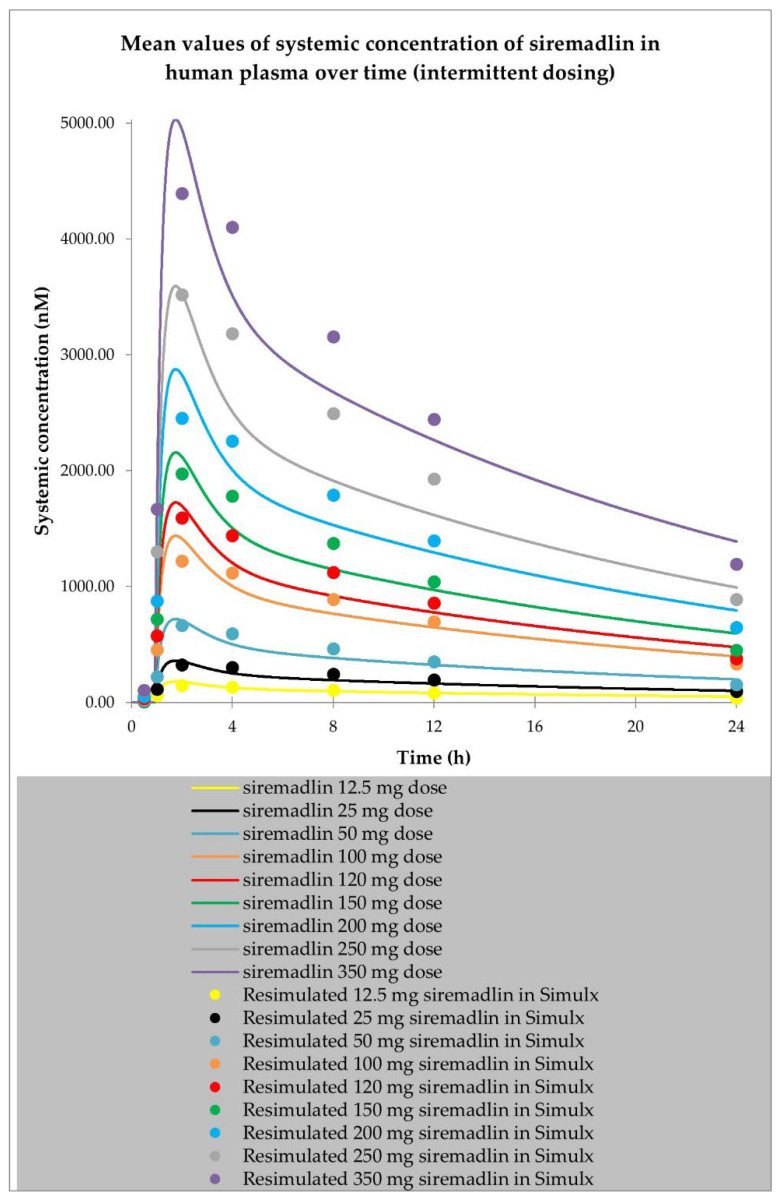
PBPK model of siremadlin administered in an intermittent regimen in cancer patient representatives using first-order absorption mechanism. Resimulated data are presented as a geometric mean from number of study participants × 10 (see Table 7 in Section 4).

**Figure 2 ijms-24-02239-f002:**
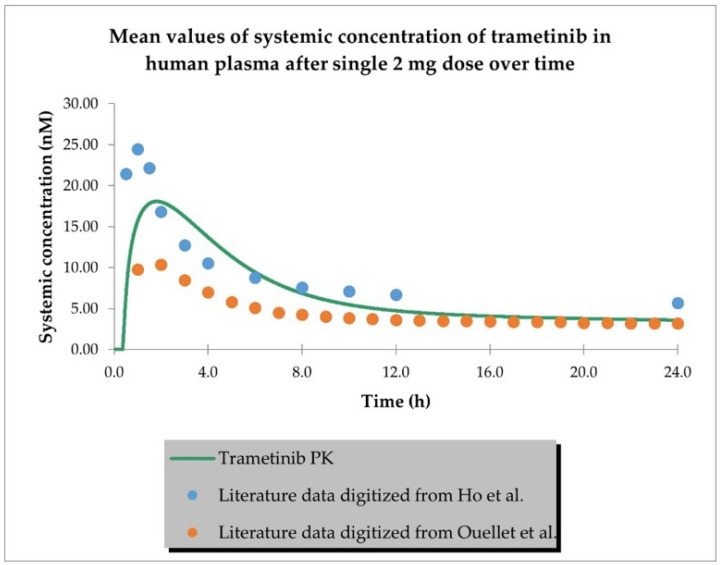
PBPK model of trametinib after a single dose in cancer patient representative. Observed data are presented as means from literature data (digitised from Ho et al. [30] and Ouellet et al. [31] as indicated in Table 3).

**Figure 3 ijms-24-02239-f003:**
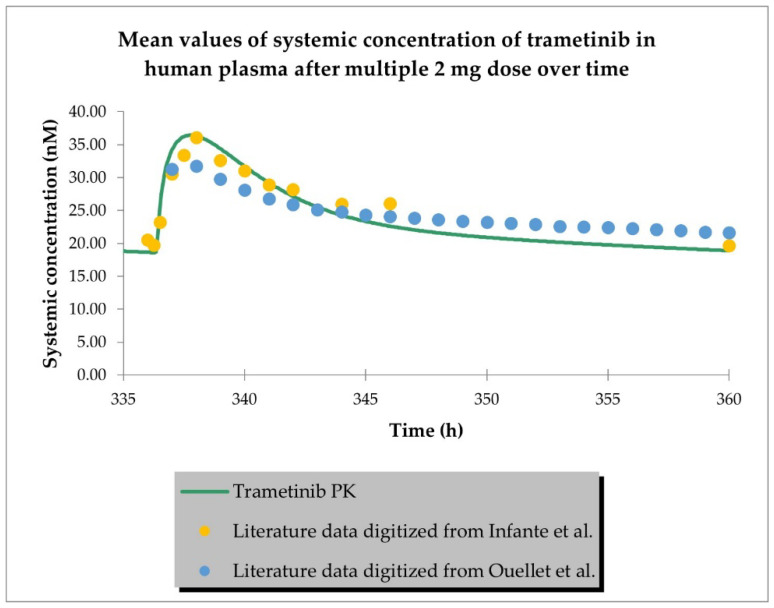
PBPK model of trametinib after a repeated dose in cancer patient representative. Observed data are presented as means from literature data (digitised from Infante et al. [32] and Ouellet et al. [31] as indicated in Table 3).

**Figure 4 ijms-24-02239-f004:**
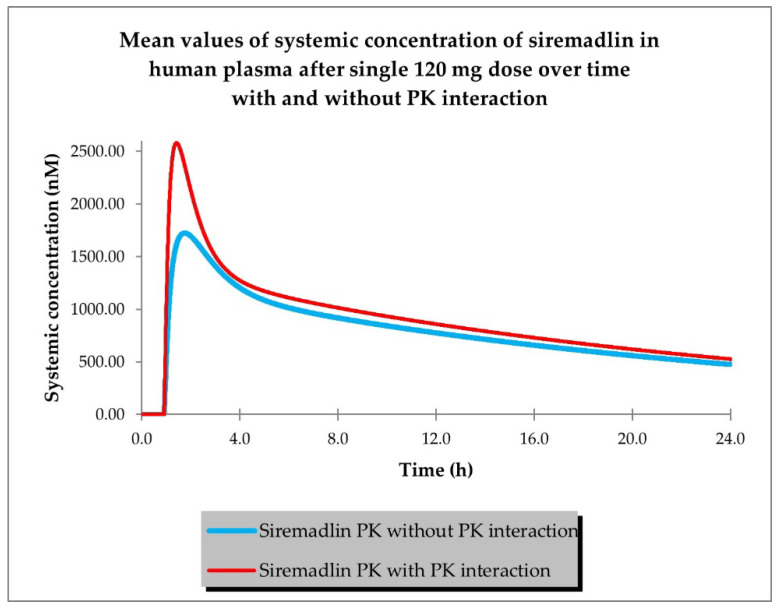
PBPK model of siremadlin with and without estimated PK interaction in cancer patient representative (*n* = 1).

**Figure 5 ijms-24-02239-f005:**
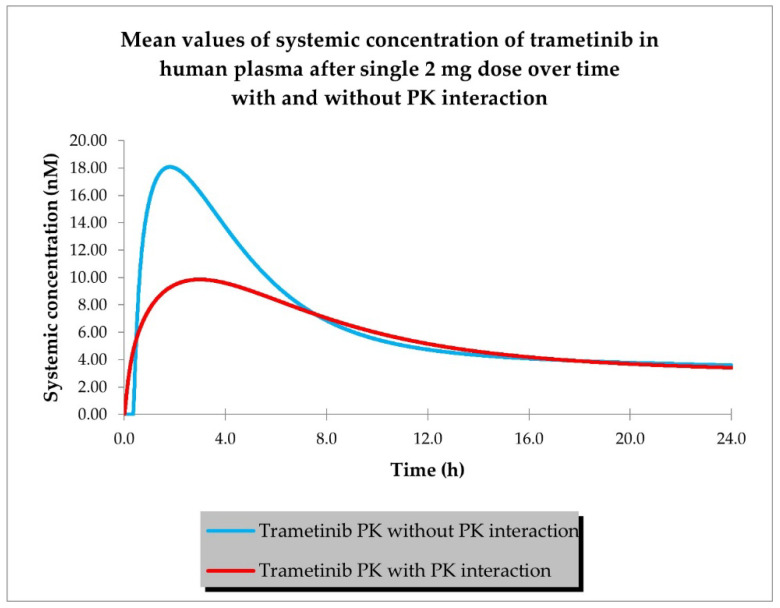
PBPK model of trametinib with and without estimated PK interaction in cancer patient representative (*n* = 1).

**Figure 6 ijms-24-02239-f006:**
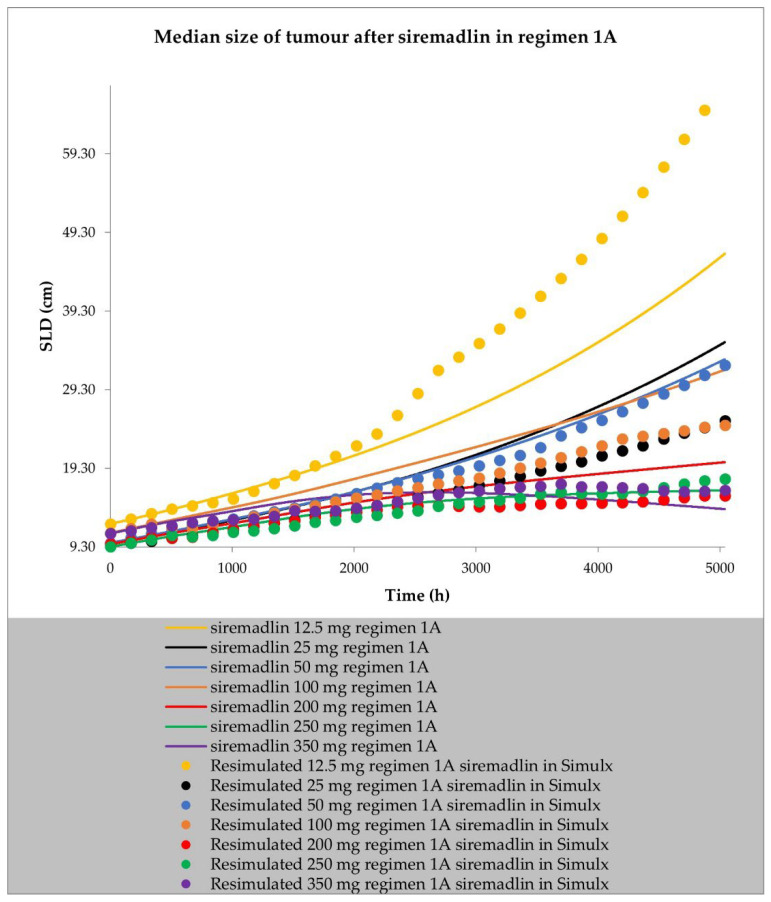
TGI model of siremadlin administered in regimen 1A in cancer patient representatives (*n* = 1 per treatment arm). Resimulated data are presented as medians from number of study participants × 10 (see Table 7 in Section 4).

**Figure 7 ijms-24-02239-f007:**
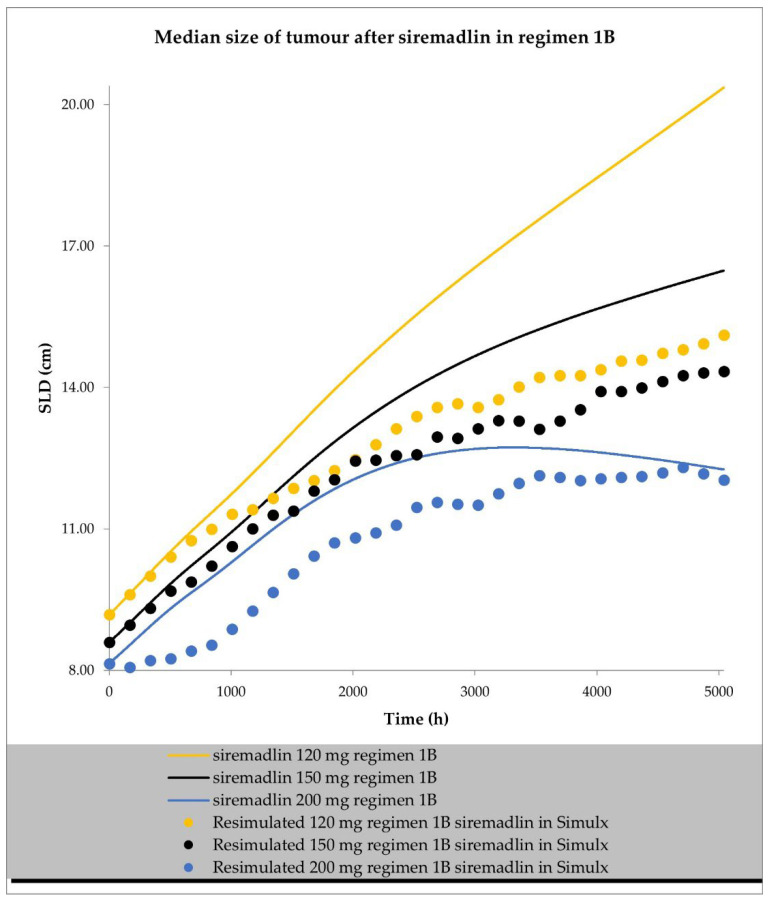
TGI model of siremadlin administered in regimen 1B in cancer patient representatives (*n* = 1 per treatment arm). Resimulated data are presented as medians from number of study participants × 10 (see Table 7 in Section 4).

**Figure 8 ijms-24-02239-f008:**
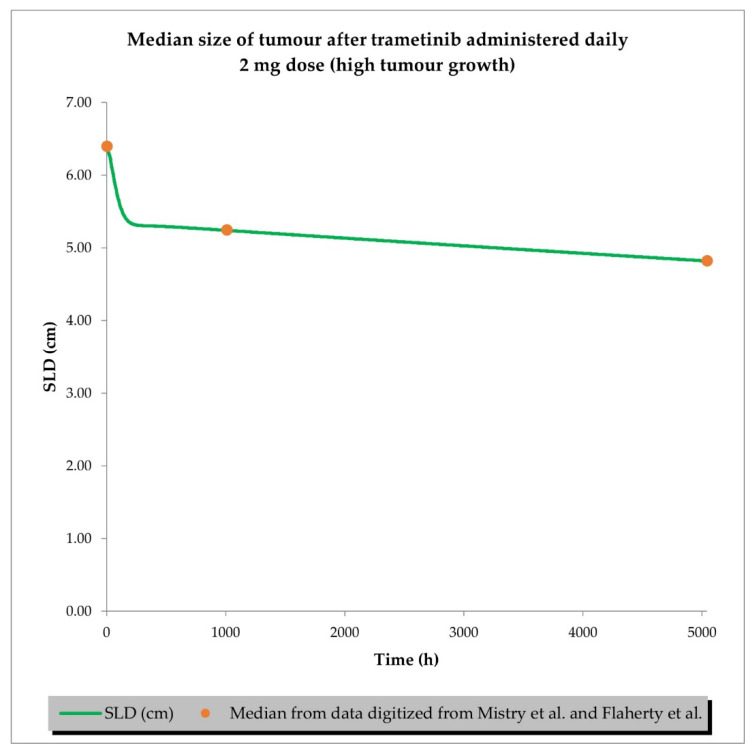
TGI model of trametinib administered daily in cancer patient representatives (*n* = 1) with assumption of high tumour growth (kgh = 0.00028 1/h). Observed data are presented as medians from literature data (data digitised from Mistry et al. [34] and Flaherty et al. [35]).

**Figure 9 ijms-24-02239-f009:**
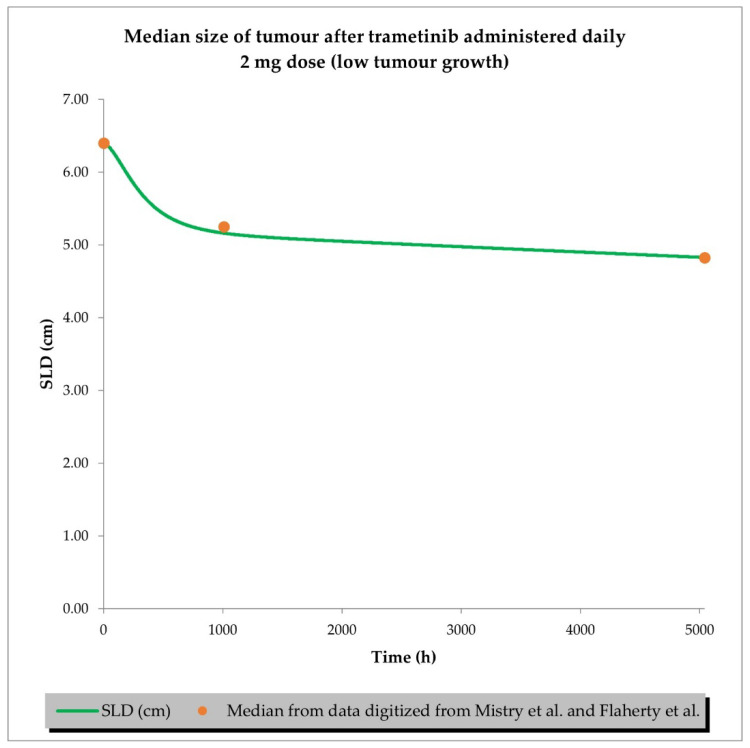
TGI model of trametinib administered daily in cancer patient representatives (*n* = 1) with assumption of low tumour growth (kgh = 0.0000261 1/h). Observed data are presented as medians from literature data (data digitised from Mistry et al. [34] and Flaherty et al. [35]).

**Figure 10 ijms-24-02239-f010:**
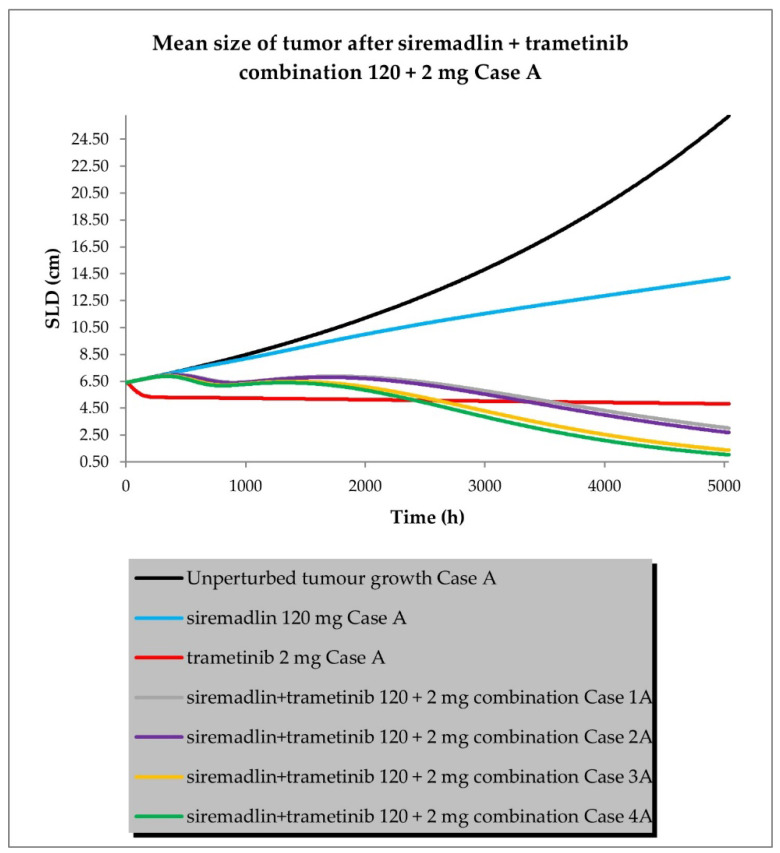
Siremadlin and trametinib combination’s 120 + 2 mg efficacy in population representative of melanoma cancer (Case A—high tumour growth, high initial tumour size, and high fraction of sensitive cells).

**Figure 11 ijms-24-02239-f011:**
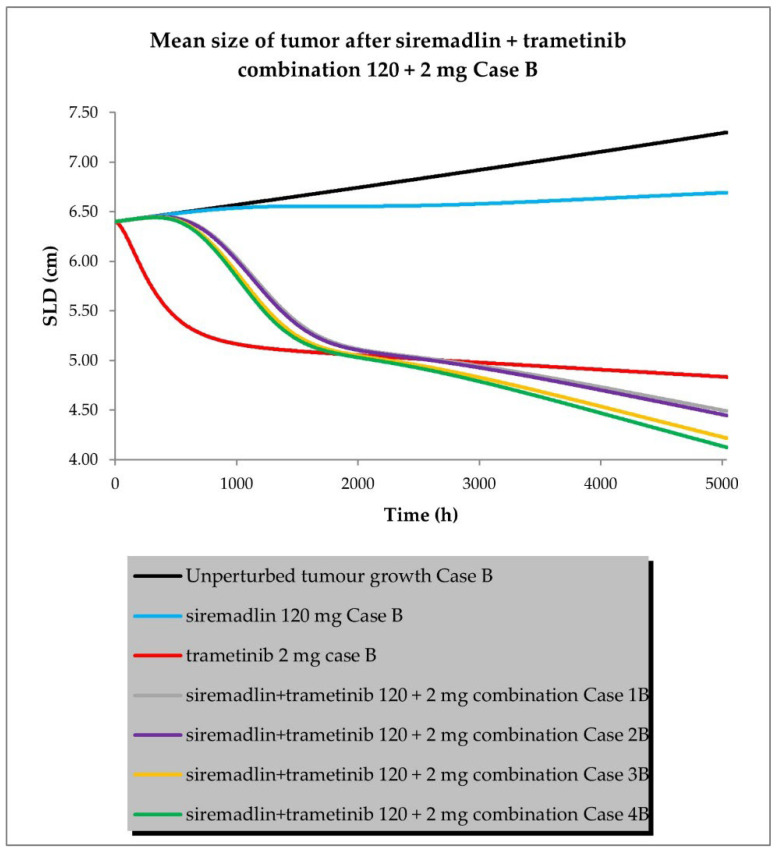
Siremadlin and trametinib combination’s 120 + 2 mg efficacy in population representative of melanoma cancer (Case B—low tumour growth, high initial tumour size, and high fraction of sensitive cells).

**Figure 12 ijms-24-02239-f012:**
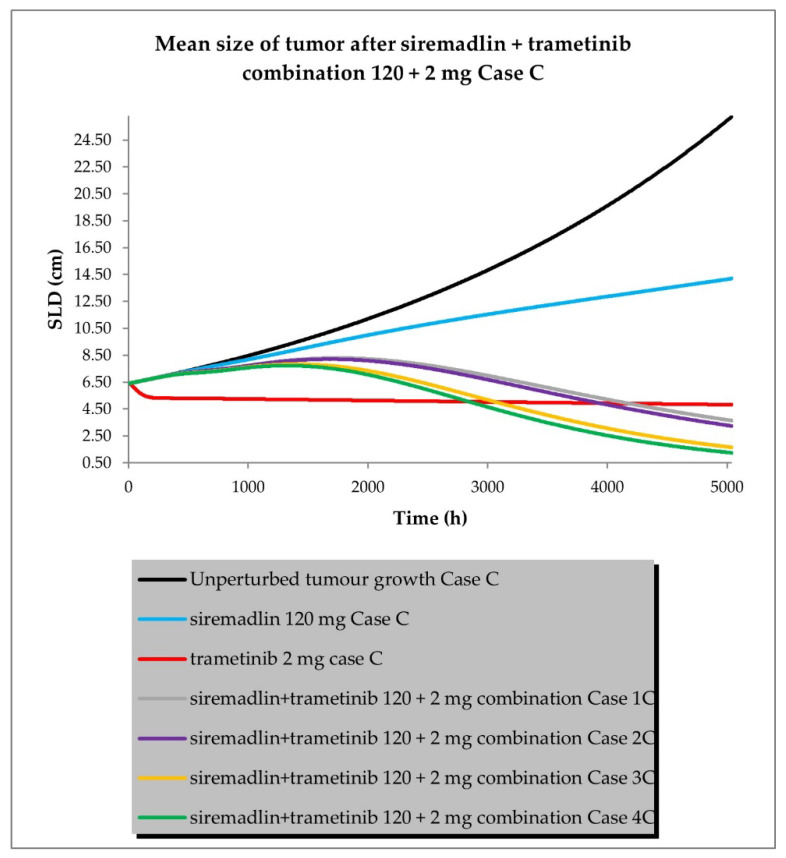
Siremadlin and trametinib combination’s 120 + 2 mg efficacy in population representative if melanoma cancer (Case C—high tumour growth, high initial tumour size, and low fraction of sensitive cells).

**Figure 13 ijms-24-02239-f013:**
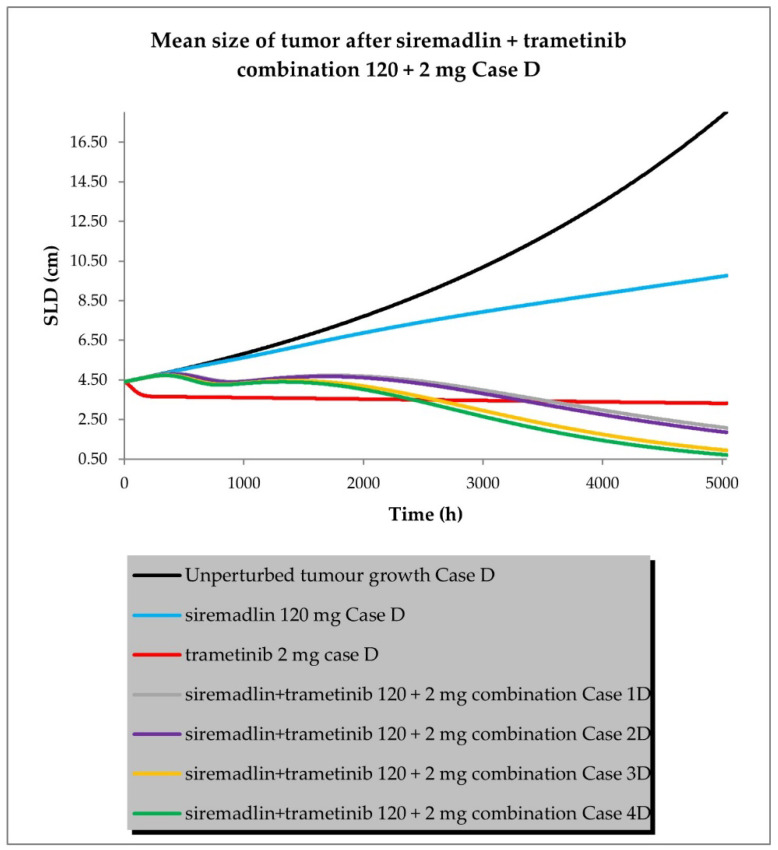
Siremadlin and trametinib combination’s 120 + 2 mg efficacy in population representative of melanoma cancer (Case D—high tumour growth, low initial tumour size, and high fraction of sensitive cells).

**Table 1 ijms-24-02239-t001:** Comparison of predicted vs. observed AUC_0-inf_ for siremadlin. Population representative and total population-derived AUC_0-inf_ parameters were generated via first-order absorption mechanism and are presented as geometric means (%CV).

Dose (mg)	Representative AUC_0-inf_ Predicted (nM × h)	Population AUC_0-inf_ Predicted (nM × h)	AUC_0-inf_ Observed (nM × h)	Representative AUC Predicted/ Observed	Population AUC Predicted/ Observed
1	257.90	286.40 (44.8%)	241.80 (-%)	1.07	1.18
2	515.80	537.83 (42.5%)	304.46 (31.5%)	1.69	1.77
4	1031.60	1073.86 (36.5%)	387.64 (22.0%)	2.66	2.77
7.5	1934.26	2013.49 (36.5%)	1076.86(49.8%)	1.80	1.87
12.5	3223.76	3580.00 (44.8%)	2670.28 (-%)	1.21	1.34
15	3868.52	3816.39 (40.0%)	2343.49 (71.0%)	1.65	1.63
20	5158.02	5119.24 (36.9%)	4122.18 (29.6%)	1.25	1.24
25	6447.53	6399.05 (36.9%)	4803.12 (28.3%)	1.34	1.33
50	12,895.05	13,423.28 (36.5%)	14,455.63 (25.6%)	0.89	0.93
100	25,790.10	26,846.56 (36.5%)	25,723.34 (58.5%)	1.00	1.04
120	30,948.13	29,889.60 (38.8%)	33,275.78 (62.7%)	0.93	0.90
150	38,685.16	37,968.46 (41.8%)	42,719.97 (43.2%)	0.91	0.89
200	51,580.21	51,192.34 (36.9%)	47,271.75 (56.2%)	1.09	1.08
250	64,475.26	63,698.83 (43.2%)	74,579.68 (71.2%)	0.86	0.85
350	90,265.34	89,586.57 (36.9%)	99,211.21 (34.4%)	0.91	0.90

**Table 2 ijms-24-02239-t002:** Comparison of predicted vs. observed C_max_ for siremadlin. Population representative and total population- derived C_max_ parameters were generated via first-order absorption mechanism and are presented as geometric means (%CV).

Dose (mg)	Representative C_max_ Predicted (nM)	Population C_max_ Predicted (nM)	C_max_ Observed (nM)	Representative C_max_ Predicted/ Observed	Population C_max_ Predicted/ Observed
1	14.37	14.05 (33.6%)	14.22 (-%)	1.01	0.99
2	28.75	28.77 (27.5%)	21.61 (23.7%)	1.33	1.33
4	57.49	58.14 (27.0%)	31.69 (22.8%)	1.81	1.83
7.5	107.79	109.01 (27.0%)	70.22 (43.9%)	1.54	1.55
12.5	179.64	175.62 (33.6%)	212.46 (-%)	0.85	0.83
15	215.60	209.79 (26.2%)	164.74 (56.9%)	1.31	1.27
20	266.36	278.51 (28.6%)	269.17 (20.3%)	0.99	1.03
25	359.32	348.15 (28.6%)	422.57 (27.5%)	0.85	0.82
50	718.63	726.77 (27.0%)	840.82 (13.0%)	0.85	0.86
100	1437.19	1453.48 (27.0%)	1194.25 (30.1%)	1.20	1.22
120	1724.76	1596.25 (26.2)	1871.59 (51.5%)	0.92	0.85
150	2156.00	2093.20 (25.4%)	2600.42 (27.8%)	0.83	0.80
200	2874.55	2785.12 (28.6%)	2104.39 (43.7%)	1.37	1.32
250	3593.23	3482.48 (27.4%)	3629.21 (69.5%)	0.99	0.96
350	5030.18	4873.72 (28.6%)	4066.91 (56.9%)	1.24	1.20

**Table 3 ijms-24-02239-t003:** Comparison of predicted vs. observed key PK parameters (AUC_0–24 h_ and C_max_) for trametinib. Population representative and population cohort-derived AUC_0–24 h_ and C_max_ parameters were generated via first-order absorption mechanism and are presented as geometric means (%CV).

Drug	Trametinib	Trametinib	Trametinib	Trametinib
Representative AUC_0–24 h_ (day 1)	165.39	-	-	-
Population AUC_0–24 h_ (day 1)	170.70 (29%)	-	200.76 (20%) *	109.60 (2%) **
Representative AUC_0–24 h_ (day 15)	570.91	-	-	-
Population AUC_0–24 h_ (day 15)	656.45 (50%)	601.24 (22%)	-	586.28 (16%) **
Population AUC ratio (day 1)	-	-	0.85	1.56
Population AUC ratio (day 15)	-	1.09	-	1.12
Representative C_max_ (day 1)	18.08	-	-	-
Population C_max_ (day 1)	18.50 (19%)	-	24.44 (3%) *	10.45 (0.3%) **
Representative C_max_ (day 15)	36.44	-	-	-
Population C_max_ (day 15)	40.93 (36%)	36.07 (28%)	-	31.67 (12%) **
Population C_max_ ratio (day 1)	-	-	0.76	1.77
Population C_max_ ratio (day 15)	-	1.13	-	1.29
Source	Current study	Digitised from Infante et al. [32]	Digitised from Ho et al. [30]	Digitised from Ouellet et al. [31]

* Geometric mean from digitised data. ** Geometric mean from digitised data from typical male and female with a median body weight of 79 kg.

**Table 4 ijms-24-02239-t004:** Comparison of predicted key PK parameters (AUC_0–24 h_ and C_max_) with and without PK interaction for population representative for siremadlin and trametinib at the 120 and 2 mg doses (with an initial tumour size of 6.4 cm).

Drug	Siremadlin	Trametinib
Dose (mg)	120	2
AUC no PK DDI	19,240.73	174.01
AUC PK DDI	21,858.34	133.15
AUC ratio (PK DDI)	1.1360	0.7652
Cmax no PK DDI	1724.96	19.02
Cmax PK DDI	2580.39	9.61
Cmax ratio (PK DDI)	1.4959	0.5051

**Table 5 ijms-24-02239-t005:** Comparison of estimated efficacy of siremadlin and trametinib combination at the 120 and 2 mg doses assuming various case scenarios.

Trial	n	No. of Trials	Difference (Drug Combination vs. Trametinib)	%ORR (Mean from 10 Trials)	Statistical Significance (*p* Value)
Siremadlin Case A	29	10	51.64%	0.00%	<0.0001
Trametinib Case A	214	10	0.00%	51.64%	<0.0001
Case 1a	243	10	26.51%	78.15%	<0.0001
Case 2a	243	10	31.16%	82.80%	<0.0001
Case 3a	243	10	45.81%	97.45%	<0.0001
Case 4a	243	10	47.29%	98.93%	<0.0001
Siremadlin Case B	29	10	25.84%	0.00%	<0.0001
Trametinib Case B	214	10	0.00%	25.84%	<0.0001
Case 1b	243	10	26.67%	52.51%	<0.0001
Case 2b	243	10	30.70%	56.54%	<0.0001
Case 3b	243	10	52.22%	78.07%	<0.0001
Case 4b	243	10	59.14%	84.98%	<0.0001
Siremadlin Case C	29	10	51.64%	0.00%	<0.0001
Trametinib Case C	214	10	0.00%	51.64%	<0.0001
Case 1c	243	10	14.41%	66.05%	<0.0001
Case 2c	243	10	20.83%	72.47%	<0.0001
Case 3c	243	10	42.52%	94.16%	<0.0001
Case 4c	243	10	45.81%	97.45%	<0.0001
Siremadlin Case D	29	10	51.64%	0.00%	<0.0001
Trametinib Case D	214	10	0.00%	51.64%	<0.0001
Case 1d	243	10	26.51%	78.15%	<0.0001
Case 2d	243	10	31.16%	82.80%	<0.0001
Case 3d	243	10	45.81%	97.45%	<0.0001
Case 4d	243	10	47.29%	98.93%	<0.0001

**Table 6 ijms-24-02239-t006:** Details of the clinical studies used for the development and verification of the siremadlin and trametinib PBPK model.

Drug	Siremadlin	Trametinib	Trametinib	Trametinib	Trametinib
NCT number	NCT02143635	NCT00687622	NCT01387204	NCT01245062	NCT00687622/ NCT01037127/ NCT01245062
Phase	1/2	1/2	1	3	1/2/3
Doses (mg)	1–350	0.125–10	2	2	0.125–10/2/2
Administration	Oral	Oral	Oral	Oral	Oral/oral/oral
n	115	206	2	214	206/97/214
Women (%)	44	46	0	44	46/30/44
Age (Years)	18–80	19–92	54–66	23–85	19–92/23–79/ 23–85
Dataset purpose	PK/PD training/ verification	PK training	PK training	PD training/ verification	PK verification
Reference	Guerreiro et al. [29] Stein et al. [36] Jeay et al. [39]	Infante et al. [32]	Ho et al. [30]	Flaherty et al. [35] Mistry et al. [34]	Ouellet et al. [31]

## Data Availability

The data presented in this study are available in the article or Appendix A. Raw data from the PK and PD simulations are available on request from the corresponding author.

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
