# Peer review of "In Vitro/In Vivo Translation of Synergistic Combination of MDM2 and MEK Inhibitors in Melanoma Using PBPK/PD Modelling: Part III"

_ijms, 2023, doi:10.3390/ijms24032239_

Round 1

Reviewer 1 Report

This paper is exhaustive, intriguing and at the same time leads towards some new ideas and directions towards treatment of melanoma, and analysing it pharmacokinetically and pharmacodynamically. To fully understand the ideas I would suggest two improvements:

1. The paper seems too lengthy or long-winded. To gather together all the main points leading to the mode of action of drugs one has to reach for facts placed far away in the text, which demands several reading of the full text. Perhaps there is a way to prepare the text more consistently. if possible.

2. Please prepare a figure of the structure of  siremadlin and trametinib. Perhaps the structures may be contained in some schemes used. Perhaps the interference with melanin production in melanoma cells may also be shown in figures.

Author Response

Thank you very much for your valuable comments and careful review of our work. Please find our point-by-point response in red down below:

  1. The paper seems too lengthy or long-winded. To gather together all the main points leading to the mode of action of drugs one has to reach for facts placed far away in the text, which demands several reading of the full text. Perhaps there is a way to prepare the text more consistently. if possible.

Thank you for this suggestion, this is a good point. In the revised version of the manuscript we:

  • shortened this work by pushing part of the non-relevant data to supplementary materials
  • expanded the description of the mode of action of siremadlin and trametinib in supplementary materials
  • added figure showing siremadlin and trametinib mechanism of action in supplementary materials (including siremadlin and trametinib structures)
  1. Please prepare a figure of the structure of siremadlin and trametinib. Perhaps the structures may be contained in some schemes used. Perhaps the interference with melanin production in melanoma cells may also be shown in figures.

That is a great suggestion. Treatment with MAPK inhibitors may increase melanin production [doi:10.3892/ijo_00000303 and 10.1073/pnas.05270769] and its increased levels may prevent melanoma metastasis [doi: 10.3892/or.2022.8432]. In the revised version of the manuscript, we included a such figure in the supplementary materials (SM). We decided to put this figure in SM instead of the introduction section in the main text to not further expand this large publication (please see it attached).

Reviewer 2 Report

The work is out of scope in my opinion. 

The authors published 2 manuscripts earlier which are based on In Vitro/In Vivo Translation (PART 1 AND 2). These two paper satisfy the scope.

For this manuscript as part 3, is totally not for special issue. A full research paper can not be published only on basis of stimulation study, and it is not bring any new information, if someone will read part 1 and part 2, there is no need for part 3. 

we built PBPK/PD models based on data from in vitro ADME, in vivo animals PK/PD and clinical data determined from the literature or estimated by the Simcyp simulator (V21).

The work is just an extension of the part 1 and 2, using some software tools. 

Another point - the scope and special doesn't match, 

the journal is not covering such topics as well. 

Author Response

Thank you very much for your valuable comments and careful review of our work. Please find our point-by-point response in red down below:

  • The authors published 2 manuscripts earlier which are based on In Vitro/In Vivo Translation (PART 1 AND 2). These two paper satisfy the scope. For this manuscript as part 3, is totally not for special issue. A full research paper can not be published only on basis of stimulation study, and it is not bring any new information, if someone will read part 1 and part 2, there is no need for part 3. 

Thank you for this comment. Part III contains new data which was obtained from already performed clinical trials. These observations we not in part II due to the publication size limit. We do not agree with the statement that part III is not bringing new information and reading part I and part II would be sufficient. Part III is showing that there is a difference between estimated clinical efficacy based on PBPK/PD models developed only on preclinical data and those built on data from clinical trials.

  • we built PBPK/PD models based on data from in vitro ADME, in vivo animals PK/PD and clinical data determined from the literature or estimated by the Simcyp simulator (V21). The work is just an extension of the part 1 and 2, using some software tools. 

Yes, we can confirm that. As intended, part III is the final chapter of this publication cycle, completing this whole project - a final piece which was not included in Part I and II.

  • Another point - the scope and special doesn't match, the journal is not covering such topics as well.

Unfortunately, we do not agree with this opinion. According to the IJMS journal website following studies are in journal scope:

  • fundamental theoretical problems of broad interest in biology, chemistry and medicine;
  • breakthrough experimental technical progress of broad interest in biology, chemistry and medicine; and
  • application of the theories and novel technologies to specific experimental studies and calculations.

In our opinion, novel technologies like for example PBPK/PD modelling and following calculations are in the scope of this journal. Moreover, at the special issue website, this scope is broadened/expanded to:
“Topics include the identification of the molecular target of new drugs, studies of drug–protein interactions, studies of the modeling and optimization of functional activity, pre-formulation studies, new pharmaceutical carrier development, and preclinical development to clinical trials.”

The publication cycle was intended to show how computational methods utilising preclinical studies findings can be incorporated into potential clinical trials which reflect the topic of preclinical development to clinical trials. Therefore, we are confident that part III of this publication cycle is in the scope of this journal and special issue.

  • Moderate English changes required

Regarding language and style, this manuscript has undergone MDPI language editing. Please find attached English language editing certificate.

Reviewer 3 Report

This manuscript describes the inhibitory effect of siremadlin, trametinib, and their combination in a clinical melanoma patient by performing PBPK/PD modelling and virtual clinical trial. They have found the most synergistic, efficacious, and safe dose levels and dosing regiments to melanoma bearing patients which is urgently needed to cure the patients.  

The experimental designing is impressive, and the manuscript is for the most part well written with substantial evidence of confirmatory and supplementary data. The discussion is also well goes with the results and postulated according to the evidence provided. The references are appropriate and timely.

Minor criticisms

• Please undergo a thorough check of the manuscript for typographical and grammatical errors.

Author Response

Dear reviewer,

Thank you very much for acknowledging our work and for your careful review of this manuscript. According to another reviewer's advice, we prepared a revised version of this manuscript which simplify this work.

Regarding your minor criticisms:

  • Please undergo a thorough check of the manuscript for typographical and grammatical errors.

Regarding language and style, this manuscript has undergone MDPI language editing. Please find attached English language editing certificate.

Round 2

Reviewer 2 Report

, author comments : we can confirm that. As intended, part III is the final chapter of this publication cycle, completing this whole project - a final piece which was not included in Part I and II.

how ever, it I snot satisfactory as it is not in scope of special issue.
